# The Metabolic Profile of Young, Watered Chickpea Plants Can Be Used as a Biomarker to Predict Seed Number under Terminal Drought

**DOI:** 10.3390/plants12112172

**Published:** 2023-05-30

**Authors:** Sarah J. Purdy, David Fuentes, Purushothaman Ramamoorthy, Christopher Nunn, Brent N. Kaiser, Andrew Merchant

**Affiliations:** 1New South Wales Department of Primary Industries, 4 Marsden Park Road, Calala, NSW 2340, Australia; 2Charles Perkins Centre, Sydney Mass Spectrometry, The University of Sydney, John Hopkins Drive, Sydney, NSW 2000, Australia; 3Plant Breeding Institute, Sydney Institute of Agriculture, School of Life and Environmental Sciences, The University of Sydney, 12656 Newell Hwy, Narrabri, NSW 2390, Australia; 4CSIRO Agriculture and Food, Australian Cotton Research Institute, 21888 Kamilaroi Hwy, Narrabri, NSW 2390, Australia; 5Sydney Institute of Agriculture, The University of Sydney, 380 Werombi Road, Sydney, NSW 2006, Australia; 6The School of Life, Earth and Environmental Science, The University of Sydney, 380 Werombi Road, Sydney, NSW 2006, Australia

**Keywords:** carbohydrates, D-pinitol, kernel weight, metabolomics, phenotype, predictive modelling, yield

## Abstract

Chickpea is the second-most-cultivated legume globally, with India and Australia being the two largest producers. In both of these locations, the crop is sown on residual summer soil moisture and left to grow on progressively depleting water content, finally maturing under terminal drought conditions. The metabolic profile of plants is commonly, correlatively associated with performance or stress responses, e.g., the accumulation of osmoprotective metabolites during cold stress. In animals and humans, metabolites are also prognostically used to predict the likelihood of an event (usually a disease) before it occurs, e.g., blood cholesterol and heart disease. We sought to discover metabolic biomarkers in chickpea that could be used to predict grain yield traits under terminal drought, from the leaf tissue of young, watered, healthy plants. The metabolic profile (GC-MS and enzyme assays) of field-grown chickpea leaves was analysed over two growing seasons, and then predictive modelling was applied to associate the most strongly correlated metabolites with the final seed number plant^−1^. Pinitol (negatively), sucrose (negatively) and GABA (positively) were significantly correlated with seed number in both years of study. The feature selection algorithm of the model selected a larger range of metabolites including carbohydrates, sugar alcohols and GABA. The correlation between the predicted seed number and actual seed number was R^2^ adj = 0.62, demonstrating that the metabolic profile could be used to predict a complex trait with a high degree of accuracy. A previously unknown association between D-pinitol and hundred-kernel weight was also discovered and may provide a single metabolic marker with which to predict large seeded chickpea varieties from new crosses. The use of metabolic biomarkers could be used by breeders to identify superior-performing genotypes before maturity is reached.

## 1. Introduction

New crop varieties with improved tolerance to climatic challenges are urgently required to enhance food security. The current processes to breed new varieties are often slow, as molecular markers are not available for many traits, so each new candidate variety must be grown to maturity before being assessed.

Metabolic biomarkers are routinely used in medicine to diagnose a condition based on its association with a specific metabolite. Examples include human chorionic gonadotropin (hCG) in urine and pregnancy, and serum creatinine in chronic kidney disease [1]. In addition to these association-based methods, medical techniques also prognostically utilise metabolic biomarkers, i.e., to predict a future outcome. An example is the prediction of cardiovascular disease, where the likelihood of a patient suffering a cardiovascular event (heart attack or stroke) is calculated using a risk prediction algorithm such as QRISK2 [2,3]. These online calculators used by general practitioners take measures of blood chemistry, along with blood pressure and physical, lifestyle and socio-economic factors, to predict the likelihood of a cardiac event within a specified time period (usually 10 years).

In plant research, metabolite profiling has frequently been used to associate the abundance of a particular metabolite with stress responses, such as increases in proline or jasmonic acid during chilling or herbivory, respectively [4,5], but more rarely has a predictive approach taken.

Metabolite profiling approaches have been used in plant studies to associate large-scale re-programming of the metabolome during treatments such as dark-induced senescence [6], cold acclimation [7] and drought response [8]. Many metabolites showed strong diurnal variation or fluctuation in abundance due to natural phenomena such as the variation in daylength or temperature [9,10,11], which has led to them being considered too variable to reliably be used to predict a trait. Indeed, the quantification of metabolites was described as being a “snapshot of a specific moment in time” [12] because of its dependence on development and the environment. However, despite the well-documented effect of the environment, it was demonstrated in a HPLC-QTOF MS study of 2475 mass peaks (noting that some metabolites result in more than one peak) that 75% were heritable and that ~800 had a heritability (H^2^) of ≥0.7 [13].

Metabolite profiling in combination with statistical methods was successfully used to predict biomass in Arabidopsis in a number of studies [14,15,16,17,18]. By combining a negative correlation with starch and a positive correlation with enzyme activities, approximately a third of the variation in the biomass of an Arabidopsis inbred family could be accounted for [16]. The statistical methods applied to generate the biomass predictive models started with pairwise associations between single metabolites and biomass using rank correlation, but it was found that the predictive power of this method was low [14,18]. Both the previous authors then used a multivariate approach to combine groups of metabolites into a model; in the case of Meyer et al. [14], this was a canonical correlation analysis (CCA), while Sulpice et al. [18] used a partial least square regression (PLS) to identify combinations of key predictors that most strongly correlated with the trait. Just two metabolites, starch and, to a lesser extent, protein, were found to be the key predictive metabolites in the study by Sulpice et al. (2009) [18]. Whilst all of these studies were performed under tightly controlled laboratory conditions for the model species, Arabidopsis, they demonstrated that the metabolic profile of plants was strongly related to phenotype, such that it could be used to predict it.

In crop species, metabolic markers for drought tolerance were identified in rice samples taken before a drought event and those taken during drought [19]. The strongest correlation was from the marker, gluconic acid, under control (non-drought-stressed) conditions with a correlation of 0.72 with shoot dry weight under drought. In maize, a positive correlation between control levels of myo-inisitol and grain yield under drought was identified [20]. These two examples are evidence that predictive markers of drought tolerance can be identified. Prognostic biomarkers for chip quality were also identified in potato, where the abundance of glucose and fructose was found to positively correlate with discolouration during frying (low chip quality). When either of these hexoses were used as markers to predict chip quality in new crosses, the correlation (RS) between predicted and measured quality was 0.67 [21]. Predictive models for yield were also developed from the biomarker profile for the bioenergy grass Miscanthus [22]. Importantly, this study was conducted under field conditions and over two years, demonstrating that a metabolic profile that correlated with performance was consistent over multiple years. However, this study also identified that many markers were not consistent, so it was important to identify only those that were robust and repeatable [22].

In studies comparing molecular versus metabolic markers, molecular markers tended to have stronger predictive power, but only slightly. In a comparison between marker types in maize, metabolic markers correlated 0.6–0.8 with the trait, whereas molecular markers were slightly higher: 0.72–0.81 [23]. However, as the authors pointed out, this meant that the 130 metabolites that were studied were almost as effective as the 38,000 molecular markers (SNPs) [23].

Chickpea is the second-most-cultivated legume in the world [24]. India and Australia were the largest global producers of chickpea, producing 9 and 2 M tonnes, respectively in 2017 [25]. In both these countries, chickpea is grown on receding soil moisture and often experiences terminal drought stress during its late reproductive growth stage. As a predominantly indeterminate species, chickpea plants continue to produce vegetative and floral biomass while water availability remains adequate [26]. The extended flowering period, combined with progressively receding water availability, results in greater reproductive inhibition as the season progresses, as drought negatively impacts pollen viability and the ability of plants to accumulate biomass [27,28]. It was estimated that up to 50% of annual global yields of chickpea are lost due to drought [29]. Yield, particularly under the influence of drought, is a complex trait that exhibits strong G × E interactions. There are no molecular markers available for this trait, so the process for yield improvement is slow. We hypothesised that metabolic biomarkers could be identified from young, healthy plants that, under field-grown conditions over two growing seasons, were (a) consistently correlated with yield traits under terminal drought and (b) strongly correlated enough to produce predictive models. Such models could be used by breeders to identify genotypes that perform well under adverse conditions, without having to expose them to the relevant stimuli (e.g., drought).

## 2. Results

### 2.1. Phenotype

Seed weight (g plant^−1^) was significantly different between years in the replicated genotypes and between rainfed and irrigated treatments in both replicated and non-replicated genotypes (Table 1, Appendix A). The hundred kernel weight (HKW) did not significantly change between years, and no significant differences were observed between treatments (Table 1). Significant differences that reflected the change in seed weight (g plant^−1^) were observed from seed number plant^−1^, which was different between years for replicated genotypes and between treatments for replicated and non-replicated genotypes (Table 1).

### 2.2. Metabolomic Profiling

Nine metabolites were profiled by gas chromatography mass spectrometry (GC-MS) from leaf tissue from the rainfed plants (prior to withholding irrigation), in both years (Table 2 and Appendix A). The most abundant metabolite in most genotypes in year 1 was malic acid, but in year 2 the sugar alcohol, D-pinitol, was in greater abundance overall than malic acid (Table 2). All the tested metabolites were significantly different in concentration between year 1 and year 2: the organic acids, such as lactic, malonic, malic and citrate, were lower in year 2, whereas the carbohydrates, including sugar alcohols, such as chiro-inositol and D-pinitol, were all in greater abundance in year 2. Exceptions were the observations regarding acids, succinate and γ-aminobutyric acid (GABA), which were also in greater abundance in year 2 (Table 2).

Carbohydrates were analysed by enzyme assays. The most-abundant carbohydrate in both years was starch, averaging 72–76 mg g^−1^ dry weight (DW) (Table 3 and Appendix A). Glucose and sucrose were in significantly greater abundance in year 2 in the replicated genotypes, but no significant differences between years were observed for either fructose or starch (Table 3).

### 2.3. Pearson’s Correlations

Pearson’s correlations were carried out to identify relationships between metabolites and seed number plant^−1^, and HKW in year 1 and year 2 (Table 4). Six out of 13 metabolites showed a significant correlation with seed number in at least one year, which were malic acid, GABA, D-pinitol, D-glucose, sucrose and starch (Table 4). Only two metabolites, sucrose and D-pinitol, showed a significant correlation with the trait in both years (*p* = <0.05), but GABA was significantly correlated in year 2, and *p* = 0.06 in year 1, so a consistent trend was observed (Table 4).

Fewer significant correlations were observed for HKW and the metabolites; indeed, only D-pinitol was significantly positively correlated in both years of testing (Table 4).

#### D-Pinitol

As the strongest Pearson’s correlation in both years was between HKW and D-pinitol, the relationship was further investigated. Both the Kabuli and Desi types were included in the year 1 trial, so the concentration in the two types was determined (Appendix A and Figure 1). The concentration of D-pinitol was significantly higher in the Kabuli types, with an average of 4.8 mg g^−1^ DW compared to 3.8 mg g^−1^ DW in the Desi types (Figure 1). The average HKW for Desi and Kabuli was 20.9 g and 40.4 g, respectively (Appendix A). Only Desi genotypes were included in the year 2 trial, but these were comprised of types of different origin, specifically, Australian breeding’ lines, Australian varieties, Indian varieties and the ICRISAT reference set (Figure 1). The concentration of D-pinitol was observed to be the highest in the Indian and Australian varieties, which were both significantly higher than those sourced from the ICRISAT reference set, while the Australian breeders’ lines were in between the two (Figure 1). The HKW values corresponded to this trend, being 19.7 g, 19 g, 14.2 g and 25 g for the four sources, respectively (Appendix A).

### 2.4. Predictive Modelling

As HKW was observed to be a largely fixed trait, not changing between years and treatments, we decided to focus the predictive modelling on seed number plant^−1^, as this was the yield parameter that showed the strongest change in response to terminal drought (Table 2). Significant correlations between seed number and multiple metabolites were observed, so a multi-variate linear regression modelling approach was employed to develop the model. The step akaike information criteria (AIC) feature selection procedure reduced the number of variables to seven, which included sucrose, D-pinitol and GABA that were identified in the Spearman’s rank correlations and also chiro-inositol, fructose, starch and the total non-structural carbohydrate (NSC) abundance (Table 5).

These selected variables were then used to train the model using a leave-one-out cross-validation (LOOCV) approach (see Materials and Methods). A strong significant correlation was observed between the predicted and actual seed number plant^−1^ with R^2^—adjusted = 0.623, demonstrating that 62% of the variation in final seed number, under terminal drought (rainfed) conditions, could be explained by seven metabolites measured from healthy, young leaves early in the growing season (Figure 2).

## 3. Discussion

We observed no significant differences in HKW between years or treatments. Complementary to our findings, was an observation that HKW did not significantly differ between chickpea genotypes under drought or watered conditions in a glasshouse trial, whereas seed number did show significant differences and also a strong correlation with yield under drought [30]. The authors concluded that HKW did not change in response to drought because, once a seed enters the phase of rapid dry weight accumulation, it has priority for assimilates over seeds in the early stage of development [30]. Hundred-kernel weight was previously shown to be the most heritable yield trait in both chickpea and broadbean, which is evidence that it is less susceptible to environmental influences [31,32]. This shows replicability in our finding that seed number, rather than size, is a major determining factor in yield under terminal drought.

Significant differences in the metabolic profile between the two years were observed, highlighting the dynamic nature of metabolites. Despite this, the correlation analyses showed that several metabolites were consistently correlated with yield traits in both years. They were sucrose, GABA (seed number) and D-pinitol (seed number and HKW). These three metabolites were all reported to increase during water stress in multiple species including Arabidopsis [33,34], rice [35], sesame [36], soybean, ricebean and other tropical legumes [37,38,39,40]. The metabolic changes that occur during stress events were linked to the ability of particular genotypes to survive or succumb. For example, a drought-tolerant variety of sorghum accumulated greater amounts of sugars and sugar alcohols during drought stress than a susceptible cultivar [41]. Similarly, in soybean, the drought tolerance of a wild accession was attributed to its capacity to accumulate a greater abundance of osmoprotective compounds during drought compared to a more susceptible line [40]. These examples link adjustments in the metabolic profile during drought (including the accumulation of carbohydrates, GABA and sugar alcohols) with improved yield performance. However, in our study, the metabolites were profiled from young, healthy, watered plants before the drought stress was imposed and a correlation between these metabolites and yield under drought was still observed. This could suggest that, to some extent, the higher yielding (high seed number) genotypes observed in our study showed a level of pre-adaptation to drought conditions. A similar observation was made in sesame, where it was reported that drought-tolerant genotypes had a higher concentration of GABA even under well-watered conditions [36]. As chickpea has been bred to complete its lifecycle under terminal drought, it appears that the metabolic adaptations that facilitate performance under these conditions have been selected and are still observable under well-watered conditions.

The positive correlation between D-pinitol and HKW and the negative relationship with seed number are results of an existing negative relationship between these two yield traits, as previously reported, which the authors attributed to parallel demands for photosynthates and nutrients [30,42]. The stronger of the two relationships was between HKW and D-pinitol.

D-pinitol is a free cyclitol that is found throughout the genus Leguminosae [39]. In mammalian systems, D-pinitol is regarded as a bioactive compound because it possesses insulin-like properties and can lower blood glucose in patients with type 2 diabetes [43,44]. In plants, the predominant association of D-pinitol is as a compatible solute, with accumulation frequently observed during abiotic stress [37,38,39]. Accumulation was observed to increase as photosynthesis declined due to drought stress, providing evidence that carbon is diverted away from the primary metabolism and into D-pinitol [39]. In transgenic tobacco that overexpressed a myo-inositol O-methyl transferase gene, *IMT1*, which catalyses the first step in the biosynthesis of the cyclic sugar alcohol D-pinitol, large quantities of ononitol accumulated [45,46]. When the transgenic plants were exposed to drought or salt treatments, they were able to retain photosynthetic performance relative to controls [46]. Therefore, D-pinitol plays a role in protecting remobilisation to the seed during filling under terminal drought conditions. The majority of scientific publications regarding D-pinitol refer to its role in stress protection (usually drought or salt) [39,46,47,48,49]. However, our results point to a more central role for D-pinitol in seed size, as consistent relationships were found, with large seeded Kabuli types having a higher concentration than smaller seeded Desi types, and Desi varieties that had presumably been selected for seed size amongst other attributes, had more than non-varieties. Of interest is the observation that ciceritol, an α-d-digalactoside of D-pinitol, accounts for 36–43% of the total sugars in chickpea seeds [50,51]. It is, therefore, possible that the increased abundance of leaf D-pinitol in larger-seeded varieties provides more of the pre-cursor material for remobilisation to the seed later in development. It would be interesting to experiment with the exogenous feeding of either D-pinitol or its pre-cursor, myo-inositol, to observe whether corresponding changes in seed size or seed ciceritol are observed.

Our results also suggest that D-pinitol could be used to identify new crosses that can produce larger seeds even before flowering occurs. For, example, if large kernel size varieties were the main aim, a cross could be made between a larger seed size parent and a parent with smaller seeds but another desirable trait (e.g., disease tolerance). The resultant progeny could be screened for leaf D-pinitol concentration before flowering occurred, and only the highest-accumulating lines could be taken forward. This would save time and money by avoiding growing plants to maturity that do not show the desirable kernel trait.

By combining a core set of markers, we were able to develop a model that could predict the number of mature seeds under drought conditions to a high degree of accuracy (R^2^ adj = 0.62). Selecting plants for abiotic stress tolerance, e.g., drought, flooding, frost and heat, is very challenging because, for large-scale breeding programs, crosses need to be screened outdoors, which is dependent upon the relevant climatic conditions occurring in a given year. The ability to predict genotypic performance under abiotic stress from non-stressed plants is of huge benefit. While, ideally, molecular markers would be more reliable for trait prediction than metabolites because they are not subject to environmental perturbations, for many crops and particularly for complex traits, they are simply not available.

Our study shows that the leaf metabolic profile of well-watered, young plants can be measured 80 days before harvest to identify, with a high degree of accuracy, which genotypes are more likely to produce higher seed numbers under terminal drought conditions. Given the close correlation between yield (g/area) and seed number plant^−1^, it is very likely that yield could also be the focus of our model. D-pinitol concentrations in the leaf are strongly and consistently associated with seed size, and this could be used as a means of early selection. The second year of our study and the year after (2018 and 2019, respectively) were the driest on record for eastern Australia. In 2021, the net value of the national welfare lost to this drought event was AUD 53 billion [52]. Extreme weather conditions, including drought, are predicted to increase in frequency and severity as part of our changing climate. Metabolite-assisted breeding offers a means to accelerate the selection of superior crosses that continue to produce viable yields under extreme climatic conditions.

## 4. Materials and Methods

### 4.1. Field Trial

The field site was located at the University of Sydney’s IA Watson Grains Research Centre, Narrabri, NSW Australia (30°16′31.7″ S, 149°48′10.7″ E). The field trial used in 2017 (year 1) was previously described [53]. The trials were sown on 5 and 7 June in year 1 (2017) and 2018 (year 2), respectively.

The field sites were 0.6 ha in total, which was divided in half, into an irrigated and rainfed treatment in an incomplete block design. For this study, the metabolite data and associated yield parameters were only collected from the rainfed side of the field. Therefore, this represents a fully randomised block design. Thirty-six genotypes were grown each year, with four replicates of each plot in each treatment (rainfed or irrigated). Plots were initially 1.6 × 6 m, which were then cut back to 4 m before podding commenced. Each plot and the perimeter of the whole trial was surrounded by a double-row of buffer plots. Seeds were planted using a five-row mechanical planter, and the row spacing was set to 0.32 m. Seeds were pre-treated with fungicide and treated with granulated inoculant (Nodulator^®^, Group-N Granular Legume Inoculant, BASF Australia Limited, Southbank, VIC, Australia) at a rate of 3.2 kg ha^−1^, and Granulock Z Extra fertiliser (Granulock^®^, Incitec Pivot Limited, Port Lincoln, SA, Australia) at a rate of 50 kg ha^−1^ was applied at the time of sowing [53].

The irrigated treatment received 25 mm irrigation (total 100 mm) approximately every two weeks from mid-August in year 1 and from May in year 2, which was homogeneously applied to the field using a lateral move irrigator. Supplementary irrigation was supplied only to the irrigated treated plots at three timepoints in year 1, but in year 2 the residual soil moisture at the start of the season was so low following the previous dry year that supplementary irrigation was supplied to both treatments until anthesis and, thereafter, only to the irrigated treatment in year 2. A total of 92 mm of irrigation was supplied during the experiment in year 1. In year 2, a total of 70 mm was applied prior to planting (in two applications), and a further 190 mm was applied to the irrigated treatment and 110 mm to the rainfed treatment over the course of the experiment. The biomarker harvests took place before drought treatment (withdrawal of irrigation) was imposed. Therefore, the “treatments” had been equally watered at the time of the biomarker leaf harvest.

### 4.2. Plant Material

Forty-nine genotypes were tested over the two years, with thirty-six included each year, and twenty-three lines being tested in both years (Appendix A). In year 1, both Desi and Kabuli types were included, but in year 2 only Desi types were cultivated. Genotypes were selected from current cultivars bred for the northern NSW region: older Australian varieties and lines sourced from ICRISAT including Indian varieties (denoted by the “ICCV” prefix) and lines from the ICRISAT reference set (denoted by the “ICC” prefix). The ICC and ICCV selections were based on pre-breeding observations and publications reporting interesting rooting/biomass/morphology and/or drought response [54,55].

### 4.3. Yield Harvest

At maturity (around day after sowing (DAS) 160), a 50 × 50 cm quadrat was placed around an area of the plot, and all plants within it were counted and then cut at the base. The plants were placed in paper bags, dried to a constant weight and then threshed to remove the seeds. Cleaned seed was weighed, and then both values were divided by the number of plants to give seed yield g plant^−1^. Hundred-kernel weight (HKW) was automated using a seed counter (Contador, Pfeuffer, Kitzingen, Germany). The average number of seeds per plant was calculated as (seed yield g plant^−1^/HKW) × 100). All plots with both treatments (rainfed and irrigated) were harvested, but only data from the rainfed plots (from which the biomarkers were harvested) were used for the model development. Machine-harvested plot yields are not included in this study because diverse genotypes were used, and the combine harvester more effectively harvested taller, larger-seeded genotypes than those with smaller seeds and stature.

### 4.4. Biomarker Harvest Protocol

Biomarker harvests took place at DAS 74 and DAS 80 in years 1 and 2, respectively. This timepoint was selected because it was the earliest that an entire stem could be harvested from each plot that would yield 20 mg dry weight of leaf material. This harvest point was when the earliest-flowering genotypes had their first emerged petals. All genotypes were at Biologische Bundesanstalt, Bundessortenamt und Chemische Industrie (BBCH) Scale 55–59. Samples were harvested from the rainfed side of the field (before the irrigation applications were ceased and drought effects took effect). Harvests were carried out on clear days between 12:00–2 pm to control for diurnal effects. A single stem that was representative of canopy height was selected from each plot, cut at the base with scissors and placed in a Whirl-Pak sample bag (Whirl-Pak, Filtration Group, https://www.whirl-pak.com/ (accessed on 15 May 2023)). Samples were frozen in liquid nitrogen, stored at −80 °C and then freeze-dried (Virtis FreezeMobile, Gardiner, MT, USA). Samples were split into leaf and stem tissues, transferred to 2 mL microcentrifuge tubes and ball-milled to a fine powder (Geno/Grinder 2010, Spex SamplePrep, Metuchen, NJ, USA). In this manuscript, only data from the leaf samples are shown.

### 4.5. Metabolite Extraction

Soluble sugars and starch were enzymatically analysed, as previously described [56,57], and GC-MS protocols were as previously described [58,59]. Metabolite extraction: Approximately 20 mg (actual weight recorded) of each freeze-dried, ball-milled plant tissue sample was weighed into 2 mL screw cap micro centrifuge tubes. Metabolites were extracted four times with 1 mL of 80% (*v*/*v*) ethanol, and the resulting supernatants were pooled; two extractions were at 80 °C for 20 min and 10 min, respectively, and the remaining two were at room temperature. A 0.5 mL aliquot of soluble metabolite extract and the remaining pellet containing the insoluble fraction (including starch) were dried down in a heat block at 50 °C until all the solvent had evaporated. The dried-down residue from the soluble fraction was then resuspended in 0.5 mL of distilled water. Samples were stored at −20 °C for analysis.

### 4.6. Soluble Sugar Analysis

Soluble sugars of samples extracted in the previous step were enzymatically quantified using a Megazyme protocol (Megazyme Sucrose, D-glucose and D-fructose Assay Procedure, K-SUFRG 04/18, Megazyme International, Co Wicklow, Ireland) by the stepwise addition of hexokinase, phosphoglucose isomerase and β-fructosidase [60]. Samples were photometrically quantified (Benchmark Plus, BioRad, Hercules, CA, USA) by measuring the change in wavelength at 340 nm for 20 min after the addition of each enzyme. Sucrose, glucose and fructose were then quantified from standard curves included on each 96-well plate.

### 4.7. Starch Quantification

Starch was quantified using a modified Megazyme protocol (Megazyme Total Starch Assay Procedure, AOAC method 996.11, Megazyme International, Co Wicklow, Ireland). Briefly, the dried pellet was resuspended in 0.4 mL of 0.2 M KOH, vortexed vigorously and heated to 90 °C in a water bath for 15 min to facilitate gelatinisation of the starch. A total of 1.28 mL of 0.15 M NaOAc (pH 3.8) was added to each tube (to neutralise the sample) before the addition of 20 µL α-amylase and 20 µL amyloglucosidase (Megazyme International, Co Wicklow Ireland). After incubation at 50 °C for 30 min and centrifugation for 5 min, a 0.02 mL aliquot was combined with 0.6 mL of GOPOD reagent (Megazyme International, Co Wicklow, Ireland). A total of 0.2 mL of this reaction was photometrically assayed (Benchmark Plus, BioRad, Hercules, CA, USA) on a 96-well microplate at 510 nm against a water-only blank. Starch was quantified from known standard curves on the same plate. Each sample and standard were tested in duplicate. Each plate contained a control sample of known concentration for both soluble sugars and starch analysis.

### 4.8. Gas Chromatography Mass Spectrometry

For the carbohydrates, sugar alcohols and organic acid analyses, gas chromatography (GC) techniques used by Merchant et al. (2006) [58] were followed accordingly. First, 50 μL of dried extract were suspended in 450 μL anhydrous pyridine, to which a solution of 1:10 ratio mixture of trimethylchloroacetamide (TMCS) and bis-trimethylsilyl-trifluroacetamide (BSTFA) was added for derivatisation. Samples were incubated for 35 min at 75 °C and analysed by GC-MS within 24 h. The analysis was carried out on an Agilent 6890 Gas Chromatograph with QQQ 7000 Mass selective detector (Agilent Technologies, Santa Clara, CA, USA). Samples were injected in a split splitless injector at 300 °C with a 20:1 split injection onto a HP-5 column (30 m, 0.25 mm ID, 0.25 μm film thickness) with helium carrier gas at a constant flow of 1 mL/min. The temperature program had an initial oven temperature set of 60 °C for 2 min, ramping to 220 °C at 10 °C min^−1^ for 5 min and then to 300 °C at 10 °C min^−1^ for 5 min. GC-MS results were identified based on retention times relative to standards and extracted ions. Peak areas were integrated, and their relative quantities were calculated by Mass Hunter software (version B.07.01, Agilent Technologies) and used for peak integration.

### 4.9. Statistics and Modelling: Linear Modelling and Feature Selection

All statistical tests, modelling and feature selection were carried out in R [61]. Student’s *t*-tests were two-sided, assuming unequal variances (*p* = <0.05).

A multivariate linear regression model was constructed to analyse the relationship between seed number plant^−1^ in rainfed (terminal drought)-treated plants and 14 metabolites analysed from well-watered conditions early in the growing season. The modelling and model evaluation and trait prediction (below) were conducted in R using the Caret (Classification and Regression Training) package [61,62]. In order to simplify the model by reducing the number of variables, the Step Akaike Information Criteria (AIC) [63,64] was applied. This maximum-likelihood estimation (MLE) feature selection technique tests whether the AIC value is increased or decreased with the step-wise addition of each explanatory variable (metabolite), with a lower value being the desired outcome. Both a forwards and backwards approach were tested, and the backwards method was found to produce the highest adjusted R^2^ value. The backwards elimination method sequentially removes variables that do not show a significant (*p* = <0.05) relationship to the trait, leaving only the minimum significantly correlated set [65,66]. A backwards approach is preferable if there is a high likelihood of collinearity amongst variables [67], which is often the case with metabolites, e.g., Ceusters et al. [68].

### 4.10. Model Evaluation and Trait Prediction

The leave-one-out cross-validation (LOOCV) approach was utilised to train the model [65]. In this approach, the model is repeatedly re-fitted using a different training and test set each time. With each iteration, a single test value (genotype) is omitted from the training set, and the mean square error (MSE) of the predicted versus actual value for that genotype is calculated. The process is repeated until all values have been used as the test value (n = 72). The test MSE is the average of all the calculated MSE’s. A linear regression between the predicted and actual values was then plotted, and the adjusted R^2^ and *p* values were determined.

## Figures and Tables

**Figure 1 plants-12-02172-f001:**
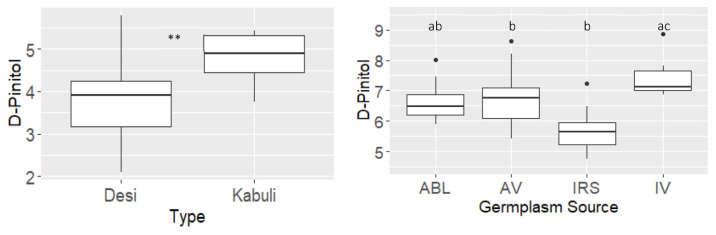
Relationships between D-pinitol and seed size. (Left) Concentration of D-pinitol (mg g^−1^ DW) in Desi (n = 29) and Kabuli (n = 7) varieties of chickpea in year 1. (Right) Concentration of D-pinitol and hundred-kernel weight (HKW) in genotypes sourced from Australian breeding lines (ABL) (n = 8), Australian varieties (AV) (n = 13), Indian varieties (IV) (n = 6) and the ICRISAT reference set (IRS) (n = 9). ** *p* = < 0.01 (Student’s *t*-test), and letters above the bars show significant differences (Tukey’s HSD Test, *p* = < 0.05).

**Figure 2 plants-12-02172-f002:**
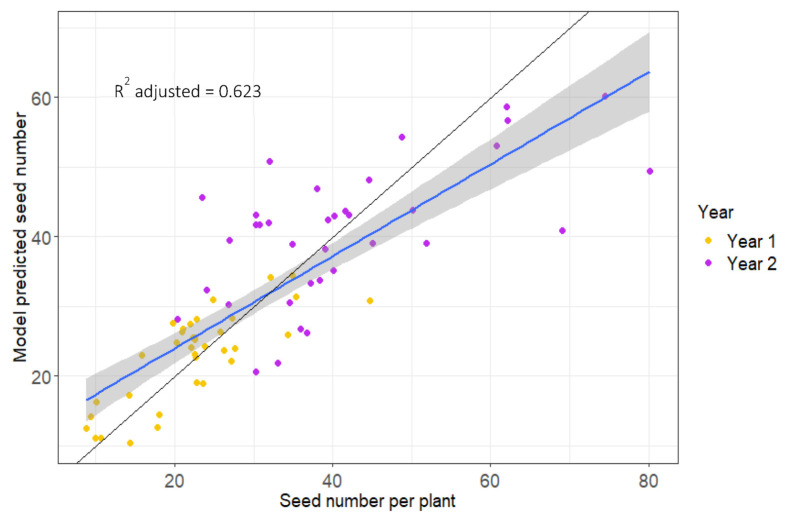
Model output showing actual seed number versus predicted seed number. Black line shows a 1:1 relationship and blue line shows linear regression between actual versus predicted seed number, grey shaded area shows confidence interval (0.95).

**Table 1 plants-12-02172-t001:** Yield parameters. Significant differences (Student’s *t*-test) beneath the values indicate significant differences between years for the replicated genotypes (** *p* = <0.01). Significant differences next to the values for replicated and unique genotypes show significant differences between the rainfed and irrigated plants (** *p* = <0.01). For replicated genotypes n = 23, and for unique genotype n = 13.

	Seed Weight (g Plant^−1^)			HKW (g)		Seed Number (Plant^−1^)		
	Rainfed	Irrigated		Rainfed	Irrigated	Rainfed	Irrigated	
Year 1 replicated genotypes	4.85	8.26	**	21.42	21.41	23.20	38.18	**
Year 2 replicated genotypes	7.16	12.74	**	21.11	22.4	34.68	58.36	**
Significantly different between years:	**	**				**	**	
Year 1 non-replicated genotypes	5.15	8.49	**	31.59	32.27	20.55	30.37	**
Year 2 non-replicated genotypes	7.63	12.73	**	15.48	17.18	53.04	77.10	**

**Table 2 plants-12-02172-t002:** GCMS metabolic profile of the leaf tissue. All values are in µg mL. n = 4.

	Lactic Acid	Malonic Acid	Succinate	Malic Acid	GABA	Citric Acid	D-Pinitol	Chiro-Inositol	Myo-Inositol
Year 1 replicated averages	0.52	0.88	0.25	8.53	0.51	3.05	3.85	0.57	0.91
Year 2 replicated averages	0.32	0.70	0.97	6.43	1.00	1.50	6.97	0.72	1.12
Significantly different between years:	**	**	**	**	**	**	**	**	**
Year 1 non-replicated averages	0.54	0.88	0.25	8.40	0.54	2.93	4.17	0.57	0.89
Year 2 non-replicated averages	0.30	0.69	0.96	6.51	1.04	1.51	5.87	0.75	1.07

GABA—γ-aminobutyric acid; Significant differences between years for the replicated genotypes are shown using a Student’s two-tailed *t*-test, ** *p* = < 0.01.

**Table 3 plants-12-02172-t003:** Enzyme assay profile of the leaf tissue. NSC = sum of non-structural carbohydrates. All values are in mg g^−1^ DW. n = 4. Significant differences between years for the replicated genotypes are shown using a Student’s two-tailed *t*-test, ** *p* = < 0.01.

	D-Glucose	D-Fructose	Sucrose	Starch	NSC
Year 1 replicated averages	8.06	8.84	21.97	76.32	115.2
Year 2 replicated averages	11.22	7.84	66.93	72.87	158.9
Significantly different between years:	**		**		**
Year 1 non-replicated averages	9.45	8.71	21.59	66.46	106.2
Year 2 non-replicated averages	10.79	7.60	60.51	72.42	151.3

**Table 4 plants-12-02172-t004:** Pearson’s correlation between seed number plant^−1^ and hundred-kernel weight (HKW), and the metabolites. Significant correlations (* *p* = < 0.05, ** *p* = < 0.01, and *** *p* = < 0.001) are coloured red and purple, and correlations where *p* = < 0.1 are coloured pink. GABA = γ-aminobutyric acid.

	Seed Number Plant^−1^				Hundred-Kernel Weight			
	Year 1		Year 2		Year 1		Year 2	
	*Rs*		*Rs*		*Rs*		*Rs*	
Lactose	0.08		0.12		−0.08		0.07	
Malonic acid	0.02		0.01		−0.04		0.21	
Succinate	0.16		0.03		−0.02		0.02	
Malic acid	0.48	***	−0.10		−0.28		−0.09	
GABA	0.31		0.34	*	−0.1		−0.16	
Citrate	0.09		−0.17		−0.2		−0.11	
D_pinitol	−0.58	***	−0.52	***	0.62	***	0.55	***
Chiro_inositol	−0.04		0.08		0.11		−0.15	
Myo_inositol	−0.07		−0.09		−0.06		0.17	
D-glucose	−0.53	***	−0.11		0.6	***	0.18	
D-fructose	−0.04		0.04		0.18		−0.02	
Sucrose	−0.39	*	−0.43	**	0.29		0.32	*
Starch	0.31		−0.15		−0.53		0.11	

**Table 5 plants-12-02172-t005:** Coefficients of significantly correlated variables.

Coefficients					
	Estimate	Std. Error	t Value	Pr	
(Intercept)	−4.81	13.75	−0.35	0.73	
GABA	46.27	8.13	5.69	0.00	***
D-pinitol	−4.64	1.35	−3.44	0.00	**
Chiro-inositol	44.03	18.26	2.41	0.02	*
D-fructose	1.74	0.61	2.85	0.01	**
Sucrose	1.37	0.38	3.57	0.00	***
Starch	1.32	0.37	3.60	0.00	***
NSC	−1.29	0.35	−3.71	0.00	***

GABA = γ-aminobutyric acid; NSC = non-structural carbohydrates; *** *p* = < 0.001, ** *p* = < 0.01, * *p* = < 0.05.

## Data Availability

All data are included in this manuscript or the supporting documents.

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
