# Peer review of "The Metabolic Profile of Young, Watered Chickpea Plants Can Be Used as a Biomarker to Predict Seed Number under Terminal Drought"

_plants, 2023, doi:10.3390/plants12112172_

Round 1

Reviewer 1 Report

In the breeding of new varieties of crops, including chickpea, it is necessary to search for effective markers predicting resistance to climate change at a very early stage of growth. More and more often, in addition to molecular markers, profiles of metabolites are used, which are often much cheaper and faster. The work entitled “The metabolic profile of young, watered chickpea plants can be used as a biomarker to predict yield traits under terminal drought” complements the knowledge on the use of metabolic markers in the prediction of yield and components. It is very puzzling why the authors do not present the results of yield assessment, but only for its component. To make full use of the presented research, such information should be included. The authors present modeling for only one of the features (HKW), justifying it by the fact that it is the most stable over the years and with or without irrigation. If the feature is not very variable, the model will also be of little information. Much more interesting is the search for a model for highly differentiated traits in which the entire genetic expression is presented. Even if no significant predictors were found for the remaining features, it is worth showing it, it will also be of interest to breeders and researchers.

Reviewer 2 Report

This manuscript is well written and shows some interesting data, but it also has severe drawbacks.

Yield is mentioned continuously throughout the manuscript, starting from the title, but no yield data are shown. I’m afraid this is inacceptable. If one wishes to seriously speak of yield traits, yield itself cannot be omitted.

The experimental setup, that is, 36 genotypes arranged in an incomplete block design with two treatments and four replicates each (irrigated and rainfed) for two years (with a total number of 49 genotypes), should be mentioned in the abstract and at the end of the Introduction, where the objectives of the study ought to be better outlined. This comment, however, must be understood as partially superseded by the remarks below.

Explorative statistics, like Principal Component Analysis, could give more clues about the biomarkers that explained most variability.

Line 191, “Table 5. Coefficients of significantly correlated variables”: as far I understand this point, these should be the coefficients of the multivariate linear regression. However, multivariate linear regression is not even mentioned in the whole “Predictive modelling” section, which is an evident omission. More details must be given in this section about the selection of the model. For example, forwards and backwards approaches are mentioned in the “Modelling: linear modelling and feature selection” section (lines 397-401), but no results about this comparison are shown. Please, amend.

In the Methods (section “Modelling: linear modelling and feature selection”) more details about the method used for curve-fitting: was partial least squares (PLS), ordinary least squares (OLS), or maximum likelihood estimation method (MLE) used?

What statistical software, and package, was used for statistical analysis in general and the multivariate linear regression model in particular?

In Table 1, what test was used to establish significant differences? It is not even mentioned in the Materials and Methods section. Please, amend.

In Tables 2 and 3, the Student’s t-test was used. First, this test does not control for the family-wise error rate at the specified alpha level when several parameters are tested. There is, therefore, a high risk of familywise Type I error, which means that the risk of falsely concluding that a treatment effect exists is too high. Kramer et al. (2016, J. Am. Soc. Hort. Sci. 141:400–406) consider using LSD (t-test) a problem for the soundness of a scientific paper in these circumstances. Second, three factors affected the dependent variables in this experiment: genotype, treatment (albeit this is not really true, as detailed below), and year. Mixed linear models can deal with highly incomplete block designs and should have been used here to provide a better overview of the various effects.

As “Biomarker harvests took place at DAS 74 and DAS 80 in years 1 and 2, respectively” (line 329), it must be specified at what stage (e.g., according to the BBCH-scale) sampling was performed. The Authors should also explain why such timing was chosen.

Lines 330-331: what does it mean “the drought treatment was imposed”? Please, clarify.

Irrigated and rainfed treatments are described as basic to this study (though this was not really true, as remarked below), and in section “Field Trial” important choices about the management of irrigation are described. How were they decided upon? Was soil moisture, or, better, soil water potential, measured? Was some stress index evaluated? These important pieces of information are missing.

Line 408, “(n=72)”: this suggests that (i)- only data for the genotypes replicated over the two years were used, and (ii)- one replicate for each genotype was used. Therefore, replication was not used at all in this experimental design. The former choice must be explained, the latter one means that the statistical design is flawed.

As “Samples were harvested from the rainfed side of the field (before the drought treatment was imposed)” (lines 330-331) and these were the samples used to predict the seed number per plant, it follows that the effect of irrigation was not accounted for in this model (as the irrigation treatment was provided only after sampling). In this way the experimental design had no real distinction between the irrigated and rainfed treatment, that is, there was no treatment at all for the predictive purposes of this study. This is a major drawback. Although the predictive model was anyway useful to some degree, the discussion of the whole study must be re-written deleting any reference to an irrigated vs rainfed treatment in the build-up of the predictive model, as the treatment effect did not really exist as a model variable. Actually, it was a confounding variable that, I suppose, caused a lot of variability in the dependent variable (yield and yield components) that could not be accounted for by the independent variable (biomarkers).

Clearly, the results about hundred kernel weight (HKW) are even more misrepresented by this severe, and obvious, drawback. The present Authors have clearly tried to disguise this aspect, which is an unfair behavior.

Given: (a)- the lack of yield data, (b)- the fact that the irrigation treatment cannot be modelled, and (c)- that there seems to be no real replication, this manuscript cannot be published in the present form.

Minor remarks are detailed below.

Lines 73-77: the year of publication is not needed. For example, after Meyer et al., (2007) [15] and Sulpice et al., (2009).

Line 313: “desi” and “kabuli” types ought to be capitalized for consistence with previous mentions.

Why “data from the leaf samples only are shown” (lines 337-338)?

Line 391: “multiVARIATE linear regression”?

Lines 411-413: it seems that the title relative to the supporting information is missing.

Reviewer 3 Report

The work is interesting. This manuscript reports on a study of The metabolic profile of young, watered chickpea plants can be used as a biomarker to predict yield traits under terminal drought. The study design meets the general standards and from what I can judge the data is being collected and analyzed appropriately. This work is an unpublished manuscript with relevant information that should be made public in a scientific journal for discussion among scientists working in the field.

However, some comments should be considered before publishing, in this way, the social and scientific relevance of the manuscript would be improved.

-        The numbering in the affiliation of the authors is incorrect. Check, filiation number 6 is missing

-        Line 20: should say: chickpeas

-        Line 23: modeling

-        Line 57: fluctuation

-        The references of lines 73 to 77 must be adjusted to the format of the journal

-        Line 88: in potatoes,

-        Line 104: 2M tons

-        Line 141: were analyzed

-        Line 160: colored

-        Line 161: colored

-        Line 231: well-watered conditions

-        Line 235: these two yields

-        Line 292: Thirty-six

-        Line 312: Forty-nine

Clear and concise writing: The abstract is well-written and clearly presents the objective, methods, results, and implications of the study.

Importance of the study: The paper highlights the importance of chickpea as a globally cultivated legume, and the challenges it faces due to terminal drought conditions. The use of metabolic profiling to predict yield traits under such conditions can potentially help farmers and breeders to develop more resilient and productive chickpea varieties.

Methods: The paper describes the methods used to measure metabolic profiles and yield parameters from field-grown chickpea leaves, as well as the predictive modeling approach used to associate metabolites with a seed number. It would be helpful if the paper provides more details on the specific metabolites measured and how they were analyzed, as well as the statistical methods used for predictive modeling (algorithm and software).

Results: The paper presents the results of the study, showing that Pinitol, Sucrose, and GABA were significantly correlated with seed number in both years of the study. The paper also reports a previously unknown association between D-pinitol and hundred kernel weight (HKW). The correlation between the predicted and actual seed number was R2 adj = 0.62, which suggests that the metabolic profile can be used to predict yield traits with a relatively high degree of accuracy.

Implications: The paper discusses the potential implications of the findings for chickpea breeding and farming practices, including the development of new varieties with higher yield potential under terminal drought conditions. The paper also suggests that the discovery of the association between D-pinitol and HKW may provide a useful tool for predicting seed size in chickpea varieties.

Line 284: I continue to add a paragraph that summarizes the importance, usefulness, and social relevance, contemporary of the study, specifically pointing out the Impact, Benefit, and Social Projection, something like this (for example):

Machine learning algorithms (ML)can process vast amounts of data and identify patterns that are difficult for humans to detect. When applied to the analysis of biomarkers in crops, ML can provide a comprehensive picture of the molecular and physiological changes that occur in response to terminal drought stress. This information can be used to predict yield characteristics and identify potential targets for crop improvement. Additionally, ML can be used to integrate data from multiple sources, such as genomics, transcriptomics, proteomics, and metabolomics, to provide a more complete understanding of the complex biological processes involved in crop response to terminal drought stress.

The use of ML to analyze biomarkers in crops under terminal drought conditions has the potential to revolutionize agricultural research and improve crop productivity in arid and tropical zones [48,49]. By identifying key biomarkers and physiological responses, ML can be used to develop crop varieties that are better adapted to terminal drought conditions, improving food security in regions where water scarcity is a major issue [50,51]. Additionally, ML can help optimize crop management strategies, such as irrigation and fertilizer application, to improve resource-use efficiency and reduce environmental impacts [52-53]. Overall, the application of ML to biomarker analysis in crops under terminal drought conditions represents a significant opportunity to address the challenges of global food security in the face of climate change.

Overall, the paper presents a well-designed study with interesting and potentially valuable findings for the chickpea industry. However, it would be helpful if the paper provides more details on the specific methods and statistical analyses used, as well as the limitations and future directions of the study.

References

I suggest adding recent references which address the issue in question. Suggested citations are for genuine scientific reasons that emphasize the current topic of study in context:

48. Vega, A.; Calderón, M.A.R.; Rey, J.C.; Lobo, D.; Gómez, J.A.; Landa, B.B. Identification of Soil Properties Associated with the Incidence of Banana Wilt Using Supervised Methods. Plants 202211, 2070. https://doi.org/10.3390/plants11152070

49. Olivares, B.; Vega, A.; Rueda Calderón, M.A.; Montenegro-Gracia, E.; Araya-Almán, M.; Marys, E. Prediction of Banana Production Using Epidemiological Parameters of Black Sigatoka: An Application with Random Forest. Sustainability 202214, 14123. https://doi.org/10.3390/su142114123

50. Paredes, F., Rey, J., Lobo, D., Galvis-Causil, S., Campos, B., The relationship between the normalized difference vegetation index, rainfall, and potential evapotranspiration in a banana plantation of Venezuela. SAINS TANAH 2021. 18(1), 58-64. http://dx.doi.org/10.20961/stjssa.v18i1.50379

51. Orlando, B.O. Tropical conditions of seasonal rain in the dry-land agriculture of Carabobo, Venezuela. Lgr 2018, 27(1):86-102. https://doi.org/10.17163/lgr.n27.2018.07

52. Campos, O.; Rey, J.C.; Perichi, G.; Lobo, D. Relationship of Microbial Activity with Soil Properties in Banana Plantations in Venezuela. Sustainability 202214, 13531. https://doi.org/10.3390/su142013531

53. Araya-Alman, M., Acevedo-Opazo, C. et al. Relationship Between Soil Properties and Banana Productivity in the Two Main Cultivation Areas in Venezuela. J Soil Sci Plant Nutr. 2020, 20 (3): 2512-2524. https://doi.org/10.1007/s42729-020-00317-8    

Round 2

Reviewer 1 Report

The work has been significantly improved and is ready for publication

Author Response

Thank you again for taking the time to review our manuscript and we thank you for supporting its publication.

Reviewer 2 Report

The Authors have now provided several essential pieces of information that were improperly omitted previously. This allows their study to be commented with better accuracy.

In my own opinion the first and foremost yield trait is, well, yield. Actually, the main reason (albeit not the only one) people look at yield components is to predict yield. This is maximally true in field experiments, and this is a field experiment. In fact, the Authors’ contention that “We do not claim to predict yield” is contradicted by the claim that “We sought to discover metabolic biomarkers in chickpea that could be used to predict grain yield” (lines 19-20). If “The paper deal specifically with seed number”, whereas HKW was not very variable, and there have been a lot of confounding issues with the yield data, then, the fairest way to deal with this is to focus the manuscript on seed number. A title like “The metabolic profile … predict the number of seeds per plant …” would be more accurate, therefore. This improved focus should be made clear throughout the manuscript.

The Authors’ model “is the relationship between multiple variables (the metabolites) harvested when the plants were young, healthy and watered, and seed number after the plants matured under terminal drought”. How can these biomarkers be predictive of seed number before any drought has happened? The only way for this to happen is if they are related to the resistance of the genotype to drought, independently of the current state of the plants. This is indeed very interesting, but it must be stated much more clearly. Moreover, this would make genotype the chief focus of this study. Thus, stating that “individual genotypes may have shown different responses in the two years, but that is not what this study aims to uncover” is totally inconsistent with the present findings. The manuscript should be extensively revised to fit this (real) focus.

Besides, as “The only one of these two traits that varied in response to drought was seed number (as shown in Table 1)”, it means that changing the seed number per plant is the chief response of chickpea to late-cycle (terminal) drought. This is true for grain legumes, in general, because of their reproductive plasticity and the overlap of vegetative and reproductive growth: differently from cereals, they continue flowering after the first seeds are set, so that, late-cycle drought blocks the development of new seeds much more than it reduces the weight of the already set grains. This is basic crop physiology and, therefore, the manuscript should focus on this aspect.

As “there were 4 replicated blocks, with a single plot of each” and “The mean of the 4 replicate plots for each year was used for the model”, the block effect was not considered in the model. This is why, I infer, a mixed model was not used. Although not a major issue, this could worsen the fit of the model. So, why did the Authors choose to avoid this kind of model? This usually happens if there are problems with the field that are not manageable with the block effect. For example, if there are troubles that are transversal (common) to the blocks. Could the Authors comment about this aspect? In the manuscript.

A further concern is that, as the data were averaged across the “4 replicate plots”, I am not convinced there were enough degrees of freedom to ensure that the effect of the studied biomarkers could be properly assessed in a linear regression model. In this respect, Harrison et al. (2018, PeerJ: e4794) argued that an ideal ratio (n/k) of data points (n) to estimated parameters (k) of 10 is appropriate, though suggestions vary widely in the literature. How many biomarkers were included in the Authors’ model (with n=72; line 443)? This must be clarified in the manuscript.

Altogether, I confirm that this manuscript shows some potentially interesting data, but I also deem that it should be thoroughly re-focused, as detailed above, according to the actual experimental design and the objective findings of this study.

Round 3

Reviewer 2 Report

I understand that “The aim of this study was to see if we could predict an outcome before an event (in the same way blood cholesterol is used as a predictor for the likelihood of a future stroke)”, but what I am saying is that, because of the experimental design used in this study, the present results are not comparable to the “way blood cholesterol is used as a predictor for the likelihood of a future stroke”: blood cholesterol can be used to predict the likelihood of a future stroke for a person. Specifically, several factors affect LDL cholesterol levels in the blood: some effects are genetic (the genotype, or ‘family history’) whereas some are not (age, smoking, obesity, unhealthy diet). In the present study, the only difference present in the experiment that can lead to a prediction of “seed number under terminal drought” - before any effect has occurred to the plants in the field - is the diversity between genotypes. Anything else that could have happened to the plants was removed by measuring biomarkers “before drought treatment (withdrawal of irrigation) was imposed” (line 238). This is a very interesting result, but it must be clarified much better. Because, contrarily to when “blood cholesterol is used as a predictor for the likelihood of a future stroke” for a given person, using the said biomarkers to predict “seed number under terminal drought” in a given field will most probably fail, because a single genotype is present in a given field, and the present biomarkers have been selected based on an experimental design that considers only the effect of genotype and not of other effects that could affect the outcome of that genotype in that field.

I wish to repeat it: the results of this study are very interesting, but they are presented in the wrong way. The present findings support that some biomarkers can individuate genetic differences that can lead to “seed number under terminal drought”, but they are not predictive of the “seed number under terminal drought” for a given field, which is what every farmer (or unspecialized scientist) would understand reading this paper in the present version.

So, I understand that “some genotypes, such as Slasher, clearly have some adaptations that predispose them to yield well under drought conditions, but exactly what those adaptations are remains to be discovered”, but I am not asking the Authors to explain what those adaptations are. I am asking to make clear that these results can show which genotype can be predicted to produce a better “seed number under terminal drought”, even though what seed number each given genotype will achieve under late-cycle drought is not predictable based on the present study. This is why the manuscript must have a different focus.

Moreover, I ask that the Authors clarify, in the Introduction, that the fact that changing the seed number per plant is the chief response of chickpea to late-cycle (terminal) drought is true for grain legumes, in general, because of their reproductive plasticity and the overlap of vegetative and reproductive growth, since, differently from cereals, they continue flowering after the first seeds are set, so that, late-cycle drought blocks the development of new seeds much more than it reduces the weight of the already set grains. This is basic crop physiology, not a finding of this study. Therefore, it must be clarified to the readers before presenting the current study.

If “There were no problems with the field trial”, then, why do the Authors refuse to show data of yield in the field? In this respect, the Authors should be aware that I can see the responses they gave to other Reviewers. The problem of the missing yield data will be evident to most readers interested in yield traits of chickpea. So, the Authors should provide a reasonable justification to explain why these data, which a reader would expect to see, are not shown. Otherwise, that reader will think that the Authors, the Reviewers, and the Academic Editor are quite naïve scientists.

Round 4

Reviewer 2 Report

I am glad that the Authors clarified any perceived misconceptions within the manuscript. I confirm that the adjustments the Authors have made to the manuscript allay all my previous concerns.